# FDG PET/CT and Endoscopic Ultrasound for Preoperative T-Staging of Esophageal Squamous Cell Carcinoma

**DOI:** 10.3390/diagnostics13193083

**Published:** 2023-09-28

**Authors:** Yung-Cheng Huang, Nan-Tsing Chiu, Hung-I Lu, Yi-Chun Chiu, Chien-Chin Hsu, Yu-Ming Wang, Shau-Hsuan Li

**Affiliations:** 1Department of Nuclear Medicine, Kaohsiung Chang Gung Memorial Hospital, Chang Gung University College of Medicine, Kaohsiung 83301, Taiwan; ychbibi@gmail.com (Y.-C.H.);; 2Department of Nuclear Medicine, National Cheng Kung University Hospital, College of Medicine, National Cheng Kung University, Tainan 704, Taiwan; 3Department of Thoracic and Cardiovascular Surgery, Kaohsiung Chang Gung Memorial Hospital, Chang Gung University College of Medicine, Kaohsiung 83301, Taiwan; 4Department of Hepato-Gastroenterology, Kaohsiung Chang Gung Memorial Hospital, Chang Gung University College of Medicine, Kaohsiung 83301, Taiwan; chiuku@ms14.hinet.net; 5Department of Radiation Oncology, Kaohsiung Chang Gung Memorial Hospital, Chang Gung University College of Medicine, Kaohsiung 83301, Taiwan; 6Department of Hematology-Oncology, Kaohsiung Chang Gung Memorial Hospital, Chang Gung University College of Medicine, Kaohsiung 83301, Taiwan

**Keywords:** esophageal squamous cell carcinoma, PET/CT, endoscopic ultrasound, staging

## Abstract

This study aimed to compare the diagnostic performances of endoscopic ultrasound (EUS) and FDG PET/CT in the preoperative T-staging of esophageal squamous cell carcinoma (ESCC) and determine whether their innovative coordination achieves better prediction. In total, 100 patients diagnosed with ESCC, 57 without (CRT^[−]sub^) and 43 with (CRT^[+]sub^) neoadjuvant chemoradiotherapy, undergoing EUS and FDG PET/CT, followed by surgical resection of the tumor, were included in this analysis. EUS classified T-stages based on the depth of primary tumor invasion, and FDG PET/CT used thresholded maximal standardized uptake value (SUV_max_) classifications. By employing pathology results as the reference standard, we assessed the accuracy of EUS and FDG PET/CT, evaluated their concordance using the κ statistic, and conducted a comparative analysis between the two modalities through McNemar’s chi-square test. FDG PET/CT had higher overall accuracy than EUS (for CRT^[−]sub^: 71.9%, κ = 0.56 vs. 56.1%, κ = 0.31, *p* = 0.06; for CRT^[+]sub^: 65.1%, κ = 0.50 vs. 18.6%, κ = 0.05, *p* < 0.01) in predicting pT- and ypT-stage. Our proposed method of incorporating both FDG PET/CT and EUS information could achieve higher accuracies in differentiating between early and locally advanced disease in the CRT^[−]sub^ group (82.5%) and determining residual viable tumor in the CRT^[+]sub^ group (83.7%) than FDG PET/CT or EUS alone. FDG PET/CT had a better diagnostic ability than EUS to predict the (y)pT-stage of ESCC. Our complementary method, which combines the advantages of both imaging modalities, can deliver higher accuracy for clinical applications of ESCC.

## 1. Introduction

Esophageal squamous cell carcinoma (ESCC), the most common histological type of esophageal cancer in Asia, is a highly aggressive malignancy with a poor survival rate, despite improvements in diagnostic methods and multimodal therapies [1]. TNM staging is critical to treatment planning and the key to individualizing therapy selection. The prognosis of patients with locally advanced disease is poor, even with aggressive surgical resection [2]. Accurate assessment of the T-stage is pivotal for predicting prognosis and guiding treatment decisions. For patients without metastasis, endoscopic or surgical curative resection alone is recommended for T1 disease [1]. For patients with operable locally advanced ESCC, such as T2 or T3 disease, esophagectomy and the dissection of lymph nodes is one of the gold standard treatment modalities for curative intent. However, solely relying on surgery for patients with locally advanced ESCC leads to unsatisfactory outcomes, with a 5-year survival rate of less than 30% [3,4,5,6,7]. To improve survival rates, a multimodal approach known as preoperative chemoradiotherapy (CRT), followed by surgery, has been recommended. This approach aims to reduce the size of the primary tumor, increasing the likelihood of successful resection while eliminating micrometastases [8,9,10]. A meta-analysis conducted by Gebski et al. reported a significant survival benefit associated with preoperative CRT in patients with operable locally advanced ESCC [11]. As a result, preoperative CRT, followed by surgery, has gradually become a common practice in many hospitals for patients with operable locally advanced T2 or T3 ESCC. For patients with inoperable locally advanced ESCC, such as T4 disease, treatment options include either preoperative CRT followed by surgery, or definitive CRT [8,9,12,13]. Hence, precise preoperative staging plays a vital role in determining the optimal treatment approach and preventing misguided attempts at curative surgery.

At present, computed tomography (CT), endoscopic ultrasound (EUS), and positron emission tomography (PET) are the prevailing techniques employed for the preoperative staging of esophageal cancer [14,15]. However, CT is unable to differentiate between the layers of the esophageal wall and is, therefore, inappropriate for assessing the T category [16]. EUS is regarded as a more accurate tool for preoperative local staging of esophageal cancer [17]. However, the efficacy of EUS in terms of determining an individual patient’s T category is inconsistent, and this method is even more deficient for assessing those receiving neoadjuvant CRT [18]. The fluorodeoxyglucose (FDG) avidity of the primary esophageal tumor is significantly positively correlated with pathological T-stage [19,20]. Moreover, PET/CT with a thresholded maximal standardized uptake value (SUV_max_) is useful for predicting the T-stage and determining residual viable tumors after CRT [21].

In this study, we extended our previous work by making direct comparisons between preoperative EUS examinations and FDG PET/CT data in subgroups of patients. We aimed to compare the diagnostic performances of EUS and FDG PET/CT with thresholded SUV_max_ in T-staging and determine ways of innovatively coordinating the two modalities.

## 2. Materials and Methods

### 2.1. Study Design

One hundred patients diagnosed with ESCC who successfully underwent both FDG PET/CT and EUS, followed by surgical resection and pathological examination of the tumor, were partially recruited from our previous study [21] for the new analysis. Patients who underwent surgery received either radical esophagectomy via video-assisted thoracoscopic surgery using either cervical esophagogastrostomy or an Ivor Lewis esophagectomy with intrathoracic anastomosis. The surgical procedure included two-field lymphadenectomy and reconstruction of the digestive tract using a gastric tube. For patients who received preoperative CRT, our approach involved administering two cycles of concurrent cisplatin and 5-fluorouracil-based chemotherapy, along with radiotherapy. The chemotherapy regimen consisted of cisplatin (75 mg/m^2^; 4-h drip) on day 1 and continuous infusion of 5-fluorouracil (1000 mg/m^2^) on days 1–4, repeated every 4 weeks. Radiotherapy was delivered in five fractions per week (50 to 50.4 Gy in 25–28 fractions), using intensity-modulated radiotherapy or three-dimensional conformal radiotherapy via a four-field technique with 6- or 10-megavolt photons. To define the treatment target, the gross target volume (GTV) was identified as the visible tumor and affected lymph nodes delineated on CT scans and/or PET/CT images; the clinical target volume (CTV) encompassed the esophagus, mediastinal, bilateral supraclavicular, and neck nodal regions; and the planning target volume (PTV) was created by expanding the CTVs with a margin of 0.5–1.0 cm in all directions. Within 4–6 weeks of completing the radiation treatment, patients underwent a series of assessments, including CT scans, endoscopy, EUS, and PET/CT, to evaluate the treatment response. After reviewing the clinical findings, the multidisciplinary team assessed the suitability of the lesions for surgical resection. If the lesions were considered resectable and the patients were deemed medically fit for esophagectomy, surgery was recommended to be performed within 6–12 weeks of the completion of CRT.

A thresholded-SUV_max_ classification, based on our previous report [21], was utilized for predicting T-stage via FDG PET/CT (fT and yfT): in patients not receiving CRT with SUV_max_ of 0–1.9: fT0; SUV_max_ of 2.0–4.4: fT1; SUV_max_ of 4.5–6.5: fT2; SUV_max_ of 6.6–13.0: fT3; SUV_max_ > 13.0: fT4; and in patients receiving CRT with SUV_max_ of 0–3.4: yfT0; SUV_max_ of 3.5–3.9: yfT1; SUV_max_ of 4.0–5.5: yfT2; SUV_max_ of 5.6–6.2: yfT3; SUV_max_ > 6.2: yfT4. According to the TNM classification, the prefix ‘‘y’’ indicates that the patient has had neoadjuvant CRT therapy before the operation. T-stages according to EUS (uT and yuT) and pathological examination (pT and ypT) were classified as the depth of the primary tumor invading the (sub)mucosa, muscularis propria, adventitia, and adjacent structures. The diagnostic performance of FDG PET/CT and EUS was calculated using pathology results as the gold standard, according to the 7th American Joint Committee on Cancer (AJCC) staging system [22]. This retrospective study was approved by our hospital’s Institutional Review Board, which waived the need to obtain consent.

### 2.2. FDG PET/CT

After the patients had fasted for at least six hours, they were given an injection of 370–555 MBq of FDG. One hour later, a PET/CT scan was performed using a combined PET/CT scanner (Discovery ST; GE Healthcare, Waukesha, WI, USA). For attenuation correction and later imaging fusion, the CT images were first acquired, without contrast medium, using the specified parameters of 40 kV, 170 mA (maximum), and 3.75-millimeter-thick sections. PET scans were then taken from mid-thigh to skull in multiple bed positions, with each scan lasting for five minutes. Transaxial PET data were reconstructed as 128 × 128-pixel images with a slice thickness of 3.27 mm using the OSEM algorithm (2 iterations, 30 subsets). Semi-quantitatively standardized uptake values (SUV) were calculated according to the following formula: SUV = measured activity within the region of interest (MBq/mL)/[injected dose of FDG (MBq)/body weight (g)]. The SUV_max_ of the esophageal tumor was then measured.

### 2.3. EUS

EUS examinations were performed using a 12.0-megahertz radial scan view with a miniprobe (GF-UM2R; Olympus, Tokyo, Japan) and a 7.5-megahertz radial scan with a conventional echoendoscope (GF-UM240; Olympus). The staging criteria classified as the depth of the primary tumor invasion were as follows: T0, no tumor seen; T1, tumor invading the mucosa or submucosa; T2, tumor invading the muscularis propria; T3, tumor invading the adventitia; and T4, tumor invading adjacent structures. The EUS examinations in the present study were performed by three gastroenterologists with expertise in endosonography, all of whom had experience of performing more than 300 EUS examinations. Tumor shapes were classified as exophytic type (polyp, protruding tumor mass, or nodule) and flat type (ulcer or uneven mucosa).

### 2.4. Statistical Analysis

Continuous variables were expressed as means with standard deviations (SD). The Kolmogorov–Smirnov test was used to test the data sets for normal distribution. Student’s *t*-test was used for group comparisons of normally distributed data, and the Mann–Whitney *U* test was used for non-normally distributed data. Categorical variables were analyzed using the chi-square test. Concordance between thresholded SUV_max_, EUS, and pathological T-stage was assessed via 5 × 5 tables using the κ statistic. The κ values were classified as follows: ≤0.2, poor agreement; 0.21–0.4, fair agreement; 0.41–0.60, moderate agreement; 0.61–0.8, good agreement; and 0.81–1, excellent agreement. The overall accuracy of FDG PET/CT and EUS was calculated using pathology results as the gold standard. McNemar’s test for data pairing by case was used to compare accuracy between groups. For the major classification, pT1, of patients who did not receive CRT, the association between SUV_max_ and the type of tumor shape and tumor size of surgical pathology specimens was further analyzed. SPSS 17 for Windows (SPSS Inc., Chicago, IL, USA) was used for all statistical analyses. Significance was set at *p* < 0.05.

## 3. Results

### 3.1. Characteristics of Patients

Of the 100 eligible patients (97 men and 3 women; mean age 54.0 ± 8.1 years) with ESCC included in this analysis, 24 patients were in the pT-stage pT0, 37 patients were in the pT1 stage, 15 patients were in the pT2 stage, 12 patients were in the pT3 stage, and 12 patients were in the pT4 stage. Primary tumors were located in the upper (*n* = 24), middle (*n* = 50), or lower esophagus (*n* = 26). The mean tumor size was 2.6 ± 1.4 cm, and the mean SUV_max_ was 5.1 ± 3.5 on FDG PET/CT. The mean intervals between PET/CT and EUS, between PET/CT and surgery, between EUS and surgery were 11.7 ± 11.8 days, 24.6 ± 19.1 days, and 31.3 ± 17.7 days, respectively. Fifty-seven patients had not undergone neoadjuvant CRT (CRT^[−]sub^ group), while 43 patients had done so (CRT^[+]sub^ group). The demographic features of the patient subgroups are summarized in Table 1.

### 3.2. CRT^[−]sub^ Group of Patients

For predicting pT-stage, the overall accuracy of FDG PET/CT with thresholded SUV_max_ was 71.9% (κ = 0.56, moderate agreement), higher than the accuracy of EUS (56.1%; κ = 0.31, fair agreement) (*p* = 0.06; Table 2). PET/CT understaged the T-stage for four patients (7.0%) and overstaged it for twelve patients (21.1%), while EUS understaged the T-stage for eleven patients (19.3%) and overstaged it for fourteen patients (24.6%). PET/CT was more accurate than EUS (78.9% vs. 75.4%; *p* = 0.77) in terms of differentiating early (pT0–1) and locally advanced (pT2–4) disease.

Eight of the twelve patients overstaged by PET/CT were in the pT1 stage. Of the 33 patients with pT1 ESCC, 12 patients had exophytic-type tumors (2 patients with polyps, 7 patients with protruding tumor masses, and 3 patients with nodules), and 21 patients had flat-type tumors (6 patients with ulcers and 15 patients with flattened or uneven mucosa). SUV_max_ was significantly higher in the exophytic type than in the flat type (7.5 ± 4.1 vs. 3.1 ± 1.0; *p* < 0.01). Compared to exophytic-type tumors, PET/CT exhibited higher accuracy in flat-type tumors (Table 3). The protruding part of the exophytic-type tumor may lead to a higher SUV value, while the T-stage is related to how deeply the tumor has grown into the esophageal wall and the surrounding tissue, not to the protruding part; therefore, we deducted the minimum value of the flat-type tumor, i.e., 2.0, from the original SUV_max_ of these exophytic-type tumors, thereby resulting in a corrected SUV_max_ and revised complementary T-stage classifications. By coordinating FDG PET/CT and EUS in this way, the accuracy of the complementary classification in terms of differentiating between early and locally advanced disease was increased from 78.9 to 82.5%. A representative case is shown in Figure 1.

### 3.3. CRT^[+]sub^ Group of Patients

For predicting ypT-stage, the overall accuracy of FDG PET/CT with thresholded SUV_max_ was 65.1% (κ = 0.50, moderate agreement), higher than the accuracy of EUS (18.6%; κ = 0.05, poor agreement) (*p* < 0.01; Table 4). This superiority was statistically significant, particularly in tumors at the middle portion of the esophagus, with small size (less than 0.2 cm), and with low SUV_max_ (less than 3.5) (Table 5). PET/CT with thresholded SUV_max_ understaged the T-stage in 10 patients (23.3%) and overstaged it in five patients (11.6%), while EUS understaged the T-stage in 11 patients (25.6%) and overstaged it in 24 patients (55.8%) in the CRT^[+]sub^ group. The accuracy of FDG PET/CT in discriminating between the residual viable tumor (non-T0) and T0 after CRT was 79.1%, which was significantly better than the 51.2% accuracy of EUS (*p* = 0.01). However, the positive predictive value of EUS for identifying residual viable tumor in yuT3–T4 (70.0%) was higher than in yuT1–T2 (35.0%). To improve the overall accuracy of the determination of the residual viable tumor, we made use of EUS for its high positive predictive value in yuT3–T4 and revised those with yfT0 but yuT3–4 to non-T0. By coordinating FDG PET/CT and EUS in this way, the accuracy of the complementary classification in terms of determining the residual viable tumor was increased from 79.1 to 83.7%. A representative case is shown in Figure 2.

## 4. Discussion

Our results showed that FDG PET/CT with thresholded SUV_max_ was more accurate than EUS (in CRT^[−]sub^ group, *p* = 0.06; in CRT^[+]sub^ group, *p* < 0.01) in predicting (y)pT-stage. A major drawback of EUS is that it is largely operator dependent and has significant interobserver variation and a long learning curve [23]. The reported accuracy of the tumor staging of EUS is highly variable, ranging from 34 to 100% (an overall accuracy of 65.55%) for esophageal adenocarcinoma [24] and from 48 to 90% (an overall accuracy of 77%) for ESCC [25]. The semi-quantitative SUV parameter measures the FDG uptake of lesions and objectively indicates metabolic activity. SUV_max_ is positively correlated with pathological T-stage in ESCC [26], non-small-cell lung cancer [27], anal carcinoma [28], and head and neck squamous-cell carcinoma [29]. We directly compared FDG PET/CT and EUS in the same group of patients and found that the former had superior diagnostic ability for predicting the (y)pT-stage of ESCC.

The accuracy of both FDG PET/CT and EUS was lower for the CRT^[+]sub^ group than the CRT^[−]sub^ group of patients in this study. EUS is considered unreliable for staging after neoadjuvant therapy [30,31,32]. The accuracy of T-staging was significantly worse after CRT (16%) than after chemotherapy (43%); the most frequent error was overstaging [33]. In this study, the accuracy of EUS in T-staging was 56.1% in the CRT^[−]sub^ group and 18.6% in the CRT^[+]sub^ group, both lower than the accuracies reported previously: 59–92% for patients without CRT [34] and 27–59% for patients with CRT [32,34,35]. Eyck et al. performed a meta-analysis of accuracy in terms of detecting residual disease after neoadjuvant CRT at the primary tumor site and showed a pooled sensitivity of 96% (range 55–100%) and a pooled specificity of only 8% (range 0–56%) [36]. The accuracy of EUS is highly affected by post-CRT inflammation and subsequent fibrosis with the distortion of the esophageal wall architecture, while these sequelae have a smaller impact on FDG PET/CT.

Compared to using the two modalities on their own, we found that further coordination of FDG PET/CT and EUS for complementary classification could achieve higher levels of accuracy in terms of differentiating between early and locally advanced disease in the CRT^[−]sub^ group and determining residual viable tumor in the CRT^[+]sub^ group. Owing to its high-quality anatomical images, EUS is useful for assessing locoregional disease. PET is not only useful for assessing distant metastases, monitoring treatment response, and restaging, but also shows sensitivity to locoregional metabolic changes. Despite its diagnostic benefits, EUS is somewhat invasive and can be restricted by severe stenoses that impede the passage of the endoscope. Moreover, it may not be adequate for some T4 tumors in locally advanced disease with a large tumor size beyond the depth of ultrasound penetration. The above limitations of EUS could be overcome via the inherently non-invasive whole-body imaging approach of PET/CT. EUS may overstage lesions with peritumoral inflammation or edema, which can be mistaken for tumor extensions, and understage tumors that microscopically infiltrate the adventitia or adjacent organs, which are beyond the resolution of the modality [37]. In the CRT^[−]sub^ group, the major erroneous classification made via FDG PET/CT was overstaging of cases, partially occurring due to the high SUV_max_ of exophytic-type tumors; this issue could be overcome by incorporating EUS information. In the CRT^[+]sub^ group, PET/CT had a higher rate of understaged cases; in contrast, EUS had a higher rate of overstaged cases. Balancing their strengths and weaknesses, the complementary classification of coordinated FDG PET/CT and EUS showed better accuracy in terms of determining residual viable tumors. Even though the improvements were not high, we achieved enhanced accuracy in terms of diagnostic applications for those important clinical usages, such as differentiating between early and locally advanced disease in the CRT^[−]sub^ group and determining residual viable tumor in the CRT^[+]sub^ group by complementarily integrating readily available non-invasive PET/CT and EUS exams.

This study has several limitations. Firstly, because of its retrospective nature, it is not feasible to re-evaluate the results of EUS examinations, which especially relied on real-time analysis and interpretation. The diagnostic differences between the three gastroenterologists who performed the examinations, although they had extensive experience, should also be considered. Nevertheless, this approach provided data closer to those obtained via routine clinical practice and may make our results more applicable. Secondly, because this study was conducted retrospectively at a single institution, there might be selection bias. The T4 cases deemed unsuitable for surgical intervention without prior preoperative CRT, which led to a small recruitment number in CRT^[−]sub^ group and reduced the utility in this category. It was difficult for us to recruit sufficient new patients directly receiving esophagectomy for analysis and validation because neoadjuvant CRT, followed by esophagectomy, has gradually become the treatment guideline for patients with operable locally advanced ESCC in our hospital, according to the large clinical trials [11,38]. Therefore, patients in this study were partly recruited from our previous study. Furthermore, indeterminate information in our dataset regarding the variability in SUV values across different PET/CT equipment and the test–retest reproducibility may limit the generalizability of our findings. Multicenter studies with larger samples and prospective designs are necessary to determine the optimal thresholds of SUV and the diagnostic value of these modalities for esophageal cancer. Therefore, while this study provides valuable insights, further studies are warranted to draw definitive conclusions.

## 5. Conclusions

In addition to its well-known usefulness in the staging of nodal and distant metastasis, FDG PET/CT has superior diagnostic ability in terms of predicting the (y)pT-stage of ESCC compared to EUS. Complementary methods that combine the advantages of both FDG PET/CT and EUS are diagnostically more accurate during clinical assessments of ESCC to differentiate between early and locally advanced disease and determine residual viable tumor.

## Figures and Tables

**Figure 1 diagnostics-13-03083-f001:**
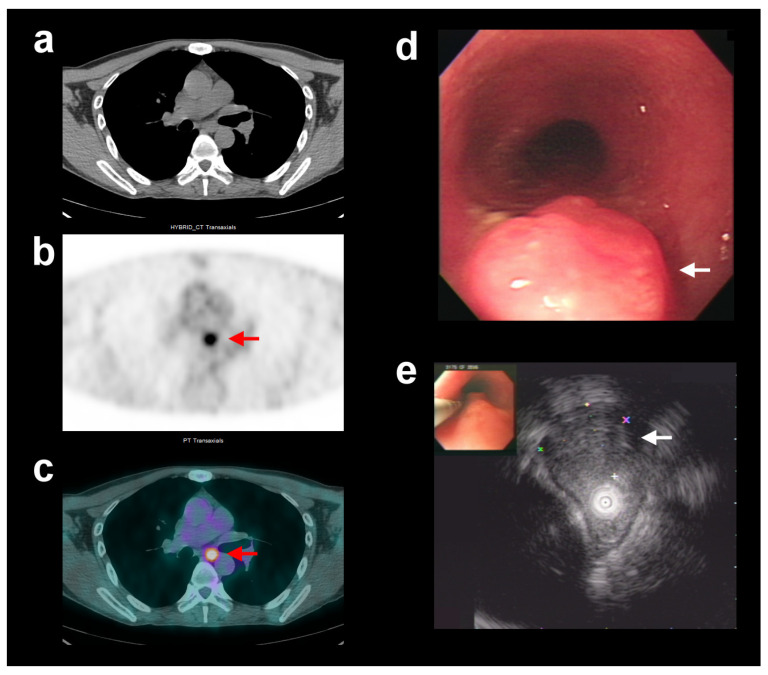
In the group of patients with esophageal squamous cell carcinoma who did not receive neoadjuvant chemoradiotherapy (CRT^[−]sub^ group), a 47-year-old man showed unexpectedly high FDG avidity in the polypoid esophageal tumor. Representative transaxial computed tomography (CT) (**a**), positron emission tomography (PET) (**b**), and fused PET/CT (**c**) images showed a focal area of increased FDG uptake in the middle thoracic esophagus (red arrows, SUV_max_ = 5.8, fT2). Endoscopic ultrasound (EUS) (**d**,**e**) showed a polypoid tumor at about the 27 cm level from central incisors with the invasion of the muscularis propria (white arrows). The application of corrected SUV_max_ for exophytic-type tumor, by coordinating FDG PET/CT and EUS, revised the complementary T-stage classification from T2 to T1, which was consistent with the final pT1 stage determined via post-surgical histopathology.

**Figure 2 diagnostics-13-03083-f002:**
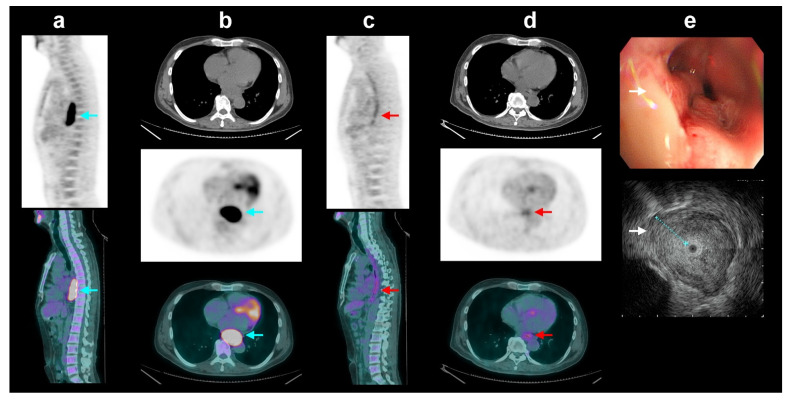
In the group of patients with esophageal squamous cell carcinoma who received neoadjuvant chemoradiotherapy (CRT^[+]sub^ group), a 52-year-old man had intense FDG avidity (blue arrows, SUV_max_ = 9.6) in the locally advanced esophageal cancer on initial sagittal (**a**) and transaxial (**b**) positron emission tomography/computed tomography (PET/CT) images (CT, PET, and fused PET/CT, top to bottom). After neoadjuvant chemoradiotherapy (CRT), the PET/CT images (**c**,**d**) showed complete metabolic response without definite residual tumor uptake (red arrows, yfT0). However, post-CRT endoscopic ultrasound (EUS) (**e**) showed an ulcerative tumor with annular wall thickening and destroyed layer structure (white arrows, maximum tumor thickness = 11.4 mm, yuT3). By coordinating FDG PET/CT and EUS, the complementary T-stage classification was revised to non-T0, making it compatible with post-surgical histopathology results.

**Table 1 diagnostics-13-03083-t001:** Demographic and clinical characteristics of patients with esophageal squamous cell carcinoma who received (CRT^[+]sub^) or did not receive (CRT^[−]sub^) neoadjuvant chemoradiotherapy.

Characteristic	CRT^[−]sub^	CRT^[+]sub^	*p*-Value
*n*	57	43	
Age, years	55.2 (8.1) ^a^	52.4 (8.0) ^a^	0.09
Sex (male:female)	55:2 (96%:4%)	42:1 (98%:2%)	0.73
Tumor location (U:M:L)	12:27:18 (21%:47%:32%)	12:23:8 (28%:53%:19%)	0.33
(y)pT-stage (T0:T1:T2:T3:T4)	4:33:10:8:2	20:4:5:4:10	<0.01
Tumor size, cm ^b^	2.6 (1.4) ^a^	2.7 (1.6) ^a^	0.68
Tumor SUV_max_	5.3 (3.7) ^a^	4.8 (3.3) ^a^	0.49
PET/CT to EUS interval, days	9.1 (9.3) ^a^	15.0 (13.8) ^a^	0.01
PET/CT to surgery interval, days	22.6 (17.5) ^a^	27.3 (21.0) ^a^	0.23
EUS to surgery interval, days	26.9 (16.7) ^a^	37.0 (17.6) ^a^	<0.01

^a^ Data are given as mean (SD); ^b^ Data are derived from measurable pathology specimens (T1–T4).

**Table 2 diagnostics-13-03083-t002:** Diagnostic performance of positron emission tomography/computed tomography (PET/CT) and endoscopic ultrasound (EUS) in patients with esophageal squamous cell carcinoma who did not receive neoadjuvant chemoradiotherapy (CRT^[−]sub^).

	Pathological T-Stage					
	pT0	pT1	pT2	pT3	pT4	
** *CRT^[−]sub^ group* **	4	33	10	8	2	
**PET/CT**						
fT0	4					
fT1		25	3	1		
fT2		6	5			
fT3		2	2	5		
fT4				2	2	
Accuracy						41/57 = 71.9%
**EUS**						
uT0	1	3				
uT1	3	21	4	1		
uT2		8	5	1		
uT3		1	1	5	2	
uT4				1		
Accuracy						32/57 = 56.1%

**Table 3 diagnostics-13-03083-t003:** Factors involved in diagnostic performance comparison analysis in patients without neoadjuvant chemoradiotherapy (CRT^[−]sub^).

Factor	Accuracy of Predict pT-Stage			Accuracy of Predict Early Disease (pT0–1)		
	PET/CT (%)	EUS (%)	*p*-Value ^b^	PET/CT (%)	EUS (%)	*p*-Value ^b^
**Tumor Location**						
upper (*n* = 12)	75.0	66.7	1.00	75.0	83.3	1.00
middle (*n* = 27)	74.1	55.6	0.23	85.2	74.1	0.45
lower (*n* = 18)	66.7	50.0	0.38	72.2	72.2	1.00
*p*-value ^a^	0.83	0.66		0.54	0.77	
**Tumor size (cm)**						
small (<2.2, *n* = 28)	71.4	60.7	0.45	75.0	75.0	1.00
large (≥2.2, *n* = 29)	72.4	51.7	0.15	82.8	75.9	0.73
*p*-value ^a^	0.93	0.49		0.47	0.94	
**Tumor SUV_max_**						
low (<4.1, *n* = 28)	85.7	67.9	0.13	85.7	82.1	1.00
high (≥4.1, *n* = 29)	58.6	44.8	0.39	72.4	69.0	1.00
*p*-value ^a^	0.02 *	0.08		0.22	0.25	
**Tumor shape**						
exophytic (*n* = 28)	53.6	42.9	0.55	67.9	67.9	1.00
flat (*n* = 29)	89.7	69.0	0.07	89.7	82.8	0.63
*p*-value ^a^	<0.01 *	0.05 *		0.04 *	0.19	

^a^ Chi-square test; ^b^ McNemar’s test; * statistically significant.

**Table 4 diagnostics-13-03083-t004:** Diagnostic performance of positron emission tomography/computed tomography (PET/CT) and endoscopic ultrasound (EUS) in patients with esophageal squamous cell carcinoma who received (CRT^[+]sub^) neoadjuvant chemoradiotherapy.

	Pathological T-Stage					
	ypT0	ypT1	ypT2	ypT3	ypT4	
** *CRT^[+]sub^ group* **	20	4	5	4	10	
**PET/CT**						
yfT0	16	3	2			
yfT1	2	1	1	2		
yfT2	1		2		1	
yfT3				1	1	
yfT4	1			1	8	
Accuracy						28/43 = 65.1%
**EUS**						
yuT0	1	1			1	
yuT1	3	1				
yuT2	10		2	2	2	
yuT3	5		2	2	5	
yuT4	1	2	1		2	
Accuracy						8/43 = 18.6%

**Table 5 diagnostics-13-03083-t005:** Factors involved in diagnostic performance comparison analysis in patients with neoadjuvant chemoradiotherapy (CRT^[+]sub^).

Factor	Accuracy of Predict ypT-Stage			Accuracy of Predict Residual Viable Tumor		
	PET/CT (%)	EUS (%)	*p*-Value ^b^	PET/CT (%)	EUS (%)	*p*-Value ^b^
**Tumor Location**						
upper (*n* = 12)	66.7	16.7	0.11	83.3	58.3	0.45
middle (*n* = 23)	65.2	8.7	<0.01 *	73.9	39.1	0.02 *
lower (*n* = 8)	62.5	50.0	1.00	87.5	75.0	1.00
*p*-value ^a^	0.98	0.04 *		0.66	0.18	
**Tumor size (cm)**						
small (<0.2, *n* = 21)	81.0	4.8	<0.01 *	81.0	9.5	<0.01 *
large (≥0.2, *n* = 22)	50.0	31.8	0.42	77.3	90.9	0.41
*p*-value ^a^	0.03 *	0.02 *		0.77	<0.01 *	
**Tumor SUV_max_**						
low (<3.5, *n* = 21)	76.2	14.3	<0.01 *	76.2	23.8	0.02 *
high (≥3.5, *n* = 22)	54.5	22.7	0.09	81.8	77.3	1.00
*p*-value ^a^	0.14	0.48		0.65	<0.01 *	

^a^ Chi-square test; ^b^ McNemar’s test; * statistically significant

## Data Availability

Data will be made available by request via correspondence.

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
