# Peer review of "FDG PET/CT and Endoscopic Ultrasound for Preoperative T-Staging of Esophageal Squamous Cell Carcinoma"

_diagnostics, 2023, doi:10.3390/diagnostics13193083_

Round 1

Reviewer 1 Report (Previous Reviewer 2)

The authors determined new thresholds of SUV in PET-CT to diagnose the depth of esophageal squamous cell carcinoma (ESCC), and reported the new thresholds showed more accurate than EUS. Additionally, they considered to combine PET-CT and EUS, resulting in better diagnostic power.

(Major problems)

・The authors determined new thresholds of SUV in previous study, and they investigated its diagnostic power comparing EUS in this study. But patients in this study are partly recruited from the previous study. Why don’t the authors recruit completely new patients for validation? In this point, the method of this study seems inappropriate.

The authors concluded that this combination of PET-CT and EUS showed better accuracy rate than PET-CT alone (82.5% vs 78.9% in non-CRT group and 83.7% vs 79.1% in CRT group). However, the difference seems to be very small. Is this significantly different?

(Minor problems)

If the authors want to describe the efficacy of combining PET-CT and EUS, they should show the table about this, like Table2 and 4.

Author Response

(Major problems)

・The authors determined new thresholds of SUV in previous study, and they investigated its diagnostic power comparing EUS in this study. But patients in this study are partly recruited from the previous study. Why don’t the authors recruit completely new patients for validation? In this point, the method of this study seems inappropriate.

Response: Thanks for your valuable comments. Indeed, it is better to recruit completely new patients for validation. Our previous study [1] recruited patients from 2007 to 2013. But, it is difficult for us to recruit new patients receiving esophagectomy directly (CRT[-]sub group: patients without neoadjuvant chemoradiotherapy) after 2013 because the treatment guideline changed in our esophageal cancer team after 2010 according to many large clinical trials. Before 2010 in our hospital, esophagectomy without neoadjuvant chemoradiotherapy was suggested for patients with 7th AJCC cT1N0M0, cT1N1M0, cT2N0M0, cT2N1M0, or cT3N0M0 esophageal squamous cell carcinoma, and neoadjuvant chemoradiotherapy followed by esophagectomy was suggested for patients with 7th AJCC cT1N2-3M0, cT2N2-3M0, or cT3N1-3M0 esophageal squamous cell carcinoma. Therefore, we can collect patients with advanced esophageal squamous cell carcinoma such as pT3 or pT4 patients in CRT[-]sub group in our previous study. However, a meta-analysis conducted by Gebski et al. in 2007 reported a significant survival benefit associated with neoadjuvant chemoradiotherapy followed by esophagectomy in patients with operable locally advanced ESCC.[2] In addition, a large phase III clinical trial,[3] the Chemoradiotherapy for Oesophageal Cancer Followed by Surgery Study (CROSS), demonstrated that neoadjuvant chemoradiotherapy followed by esophagectomy has a significant survival benefit compared to esophagectomy alone. Thus, neoadjuvant chemoradiotherapy followed by esophagectomy was gradually applied to clinical practice in many hospitals for patients with operable locally advanced esophageal squamous cell carcinoma. Therefore, after 2010, our esophageal cancer team in our hospital modify our treatment guideline as follows. Esophagectomy without neoadjuvant chemoradiotherapy was suggested for patients with 7th AJCC cT1bN0M0 esophageal squamous cell carcinoma, and neoadjuvant chemoradiotherapy followed by esophagectomy was suggested for patients with 7th AJCC cT1N1M0, cT1N2-3M0, cT2N0M0, cT2N1M0, cT2N2-3M0, or cT3N0-3M0 esophageal squamous cell carcinoma. For patients with cT1aN0M0 esophagecl squamous cell carcinoma, esophageal submucosal dissection or esophageal mucosal resection was suggested. Therefore, after 2013, there were very few patients receiving esophagectomy directly without neoadjuvant chemoradiotherapy. It is difficult for us to recruit new patients receiving esophagectomy directly (CRT[-]sub group) for validation. Therefore, patients in this study are partly recruited from the previous study. Sorry for our inadequacy and thanks for your understanding.

Reference

  1. Huang, Y.C.; Lu, H.I.; Huang, S.C.; Hsu, C.C.; Chiu, N.T.; Wang, Y.M.; Chiu, Y.C.; Li, S.H. FDG PET using SUV(max) for preoperative T-staging of esophageal squamous cell carcinoma with and without neoadjuvant chemoradiotherapy. BMC Med Imaging 2017, 17, 1, doi:10.1186/s12880-016-0171-7.
  2. Gebski, V.; Burmeister, B.; Smithers, B.M.; Foo, K.; Zalcberg, J.; Simes, J. Survival benefits from neoadjuvant chemoradiotherapy or chemotherapy in oesophageal carcinoma: a meta-analysis. Lancet Oncol 2007, 8, 226-234, doi:S1470-2045(07)70039-6 [pii]10.1016/S1470-2045(07)70039-6.
  3. van Hagen, P.; Hulshof, M.C.; van Lanschot, J.J.; Steyerberg, E.W.; van Berge Henegouwen, M.I.; Wijnhoven, B.P.; Richel, D.J.; Nieuwenhuijzen, G.A.; Hospers, G.A.; Bonenkamp, J.J., et al. Preoperative chemoradiotherapy for esophageal or junctional cancer. N Engl J Med 2012, 366, 2074-2084, doi:10.1056/NEJMoa1112088.

・The authors concluded that this combination of PET-CT and EUS showed better accuracy rate than PET-CT alone (82.5% vs 78.9% in non-CRT group and 83.7% vs 79.1% in CRT group). However, the difference seems to be very small. Is this significantly different?

Response: Combination of PET/CT and EUS showed no definite difference in the overall accuracy for predicting the pT-stage, that is the reason we omitted the extended Table 2 and 4 for combination data as the “minor problems” mentioned below. However, by complementary integrating readily available non-invasive PET/CT and EUS exams, we aimed to enhance the accuracy of diagnostic applications for those important clinical usages (differentiating between early and locally advanced disease in the CRT [-]sub group and determining residual viable tumor in the CRT[+]sub group), even if the improvement is not high. 

(Minor problems)

・If the authors want to describe the efficacy of combining PET-CT and EUS, they should show the table about this, like Table2 and 4.

Response: We omitted the presentation of extended Table 2 and 4 for combination data because we wanted to emphasize clinical applications rather than improving diagnostic accuracy for predicting the pT-stage in these fields. However, if the reviewer consider that showing it would be valuable, we are more than willing to present the extended combination data.

Reviewer 2 Report (Previous Reviewer 3)

I can't confirm the previous review, but I think there is no problem because it seems that the points I pointed out have been corrected.

English was well written.

Author Response

Thank you again for your valuable time.

Round 2

Reviewer 1 Report (Previous Reviewer 2)

Thank you for replying my comments.

1. Now I understand it is difficult to recruit new patients for CRT(-) group. Can you describe about this point as a limitation?

2. I understand the authors’ opinion about the combination of PET-CT and EUS. Can you describe about the significance of combining PET-CT and EUS (i.e. importance of differentiating between early and locally advanced disease in the CRT [-]sub group and determining residual viable tumor in the CRT[+]sub group) in discussion?

Author Response

Dear reviewers and editors:

  We thank the reviewers and editors for their thorough review of our response, and their valuable and constructive comments. We revised our manuscript according to the referees’ comments and upload the revised file (content-R.docx). The revisions to the manuscript were highlighted

Reviewer 1:

  1. Now I understand it is difficult to recruit new patients for CRT(-) group. Can you describe about this point as a limitation?

Add in the Discussion section, paragraph 4 (limitation), line 305:

It was difficult for us to recruit sufficient new patients receiving esophagectomy directly for analysis and validation because neoadjuvant CRT followed by esophagectomy has gradually become the treatment guideline for patients with operable locally advanced ESCC in our hospital according to the large clinical trials [38,39]. Therefore, patients in this study were partly recruited from our previous study.

Reference

  1. Gebski V; Burmeister B; Smithers BM; Foo K; Zalcberg J; Simes J, et al. Survival benefits from neoadjuvant chemoradiotherapy or chemotherapy in oesophageal carcinoma: a meta-analysis. Lancet Oncol. 2007, 8, 226-234.
  2. van Hagen P; Hulshof MC; van Lanschot JJ; Steyerberg EW; van Berge Henegouwen MI; Wijnhoven BP, et al. Preoperative chemoradiotherapy for esophageal or junctional cancer. N Engl J Med. 2012, 366, 2074-2084.

  1. I understand the authors’ opinion about the combination of PET-CT and EUS. Can you describe about the significance of combining PET-CT and EUS (i.e. importance of differentiating between early and locally advanced disease in the CRT [-]sub group and determining residual viable tumor in the CRT[+]sub group) in discussion?

Add in the Discussion section, paragraph 3, line 291:

Even though the improvements were not high, we achieved enhanced accuracy in diagnostic applications for those important clinical usages, such as differentiating between early and locally advanced disease in the CRT[-]sub group and determining residual viable tumor in the CRT[+]sub group by complementarily integrating readily available non-invasive PET/CT and EUS exams.

This manuscript is a resubmission of an earlier submission. The following is a list of the peer review reports and author responses from that submission.

Round 1

Reviewer 1 Report

This study evaluated the correlation between clinical T staging before surgery and pathological T staging for locally advanced ESCC. In this field, T4 or not is the most critical problem. But this study did not clear this problem because of a few T4 cases.

Nothing.

Author Response

RESPONSE:

Not only early ESCC, we also included locally advanced ESCC for this study. Because of T4 lesion was not suitable for surgery without preoperative CRT in clinical practice, there was relatively less patient with a T4 lesion in the CRT[−]sub group. Indeed, this is one of limitation due to the nature of this retrospective study. We addressed this point in the “limitation” paragraph in discussion section.

Revised the Discussion section, paragraph 4.

Second, because this study was conducted retrospectively at a single institution, there might be selection bias. The T4 cases deemed unsuitable for surgical intervention without prior preoperative CRT, which led to a small recruitment number in CRT[-]sub group and reduced the utility in this category. Furthermore, indeterminate information in our dataset regarding the variability in SUV values across different PET/CT equipment and the test-retest reproducibility may limit the generalizability of our findings. Multicenter studies with larger samples and prospective designs are necessary to determine the optimal thresholds of SUV and diagnostic value of these modalities for esophageal cancer. Therefore, while this study provides valuable insights, further studies are warranted to draw definitive conclusions.

Reviewer 2 Report

The authors investigated the accuracy of diagnosis using FDG-PET and EUS. They concluded FDG-PET showed better diagnostic power than EUS. Even after CRT, FDG-PET showed better diagnostic power.

(Minor problem)

The thresholds of SUVmax for diagnosing the tumor depth were determined by the almost same patients group in this study. Therefore, it is no wonder that these thresholds showed better accuracy. Do the authors think these thresholds are useful in other patients with ESCC? The authors should describe their idea about this point.

Author Response

RESPONSE:

The SUV of lesions were all derived from one individual PET/CT equipment in a single institution, which offered consistency of our measurement. However, the variance in SUV values of different PET/CT equipment and the test-retest reproducibility were unavailable from these data. Our results should be validated in prospective multicenter trials of FDG PET/CT for esophageal cancer.

Revised the Discussion section, paragraph 4.

Second, because this study was conducted retrospectively at a single institution, there may be selection bias. The T4 cases deemed unsuitable for surgical intervention without prior preoperative CRT, which led to a small recruitment number in CRT[-]sub group and reduced the utility in this category. Furthermore, indeterminate information in our dataset regarding the variability in SUV values across different PET/CT equipment and the test-retest reproducibility may limit the generalizability of our findings. Multicenter studies with larger samples and prospective designs are necessary to determine the optimal thresholds of SUV and diagnostic value of these modalities for esophageal cancer. Therefore, while this study provides valuable insights, further studies are warranted to draw definitive conclusions.

Reviewer 3 Report

This study is interesting, but there is one point to revise before publishing.

Could you compare 2 group’s characteristics and do propensity score matching, if you consider in this study both groups who receive or not CRT?

This article is written well.

Author Response

RESPONSE:

By employing pathology results as the reference standard, we compared the diagnostic performance of EUS and FDG PET/CT in preoperative T-staging of ESCC in the two independent groups, separately. Due to the different nature of CRT[-]sub and CRT[+]sub patients, we used different thresholds of SUV for the two groups. Every patient had both EUS and PET/CT data. We didn’t intend to compare diagnostic performance cross the two group and we didn’t have experiment and control group. Thus, we ask to omit the propensity score matching. If the group’s characteristics comparison is acceptable, according the suggestion, we could add the p-value in table 1.

Revised the table 1.
